# The role of International Civil Society Organizations in democratization: A crisp-set QCA approach to anti-corruption in Ghana

Ebenezer Kurtis Graham 🄳 *◉, Özker Kocadal◉

Department of Political Science, and International Relations, Cyprus International University, Nicosia, Cyprus

◉ These authors contributed equally to this work.
* ogyando@gmail.com

## Abstract

An anti-corruption strategy is essential in the process of continuous democratisation for effective governance. The objective of our study is to examine the mechanisms that contribute to an effective anticorruption strategy in a case study of civil society organisations in Ghana. These CSO cases illustrate the efforts of non-state actors to promote democracy in their interactions with Ghana's government. A model of anticorruption strategy and the crisp-set qualitative comparative analysis (cs-QCA) method were employed to examine 264 cases of local civil societies in Ghana that responded to questions on the conditions for an effective anticorruption strategy. Our results indicate that no single condition is sufficient as an anti-corruption strategy. However, the role of international civil society organisations (RISCO) is necessary for any anticorruption strategy. RISCO has to be combined with freedom of expression, good leadership, fear of punishment, and training to provide an effective strategy. On the other hand, despite the fear of punishment, social trust and leadership, rules and regulations, and training, the absence of RISCO explains all ineffective anticorruption strategies. Therefore, Ghana's anti-corruption strategy must always consider the role of organisations such as Amnesty International (AI), Transparency International (TI), and the Global Organization of Parliamentarians against Corruption (GOPAC) in mitigating corruption. Policymakers should therefore promote the presence of international civil society in Ghana because they ultimately contribute to democratization in addition to all government effort.

## 1. Introduction

Democracy is commonly viewed as a prerequisite for political modernisation and maturity [1]. However, democratisation involves the consolidation of democratic practices. While theories of democracy have discussed the requirements for democratisation at the macro level, such as the change in government from authoritarian to democratic regimes, civil society plays a significant role in the process. Civil society organisations (CSO) have made significant contributions to the promotion of democratic culture globally. However, Ghana's democratisation process is still ongoing, despite being a nation in Huntington's third democratic wave. Several

**Data Availability Statement:** The data used in this study is accessible via Kaggle database. https://www.kaggle.com/datasets/ebenezerkurtisgraham/a-csqca-approach-to-anti-corruption-in-ghana.

**Funding:** The author(s) received no specific funding for this work.

**Competing interests:** The authors have declared that no competing interests exist.

military *coups*, *d'état*, and other misconduct in democratic processes have led to the back-sliding of democracy in Ghana. Within democratization studies, the role of civil society has therefore been increasingly examined being the third sector [2, 3].

## 1.1 The relevance of Civil Society Organization in the domestic politics

Colonial rule and the pattern of power transfer have made African countries unprepared for democratic statehood because democratic representation procedures and liberal values are foreign ideas. Furthermore, the formation of responsible and democratic citizens was impeded by tendencies toward authoritarian practices and challenges to national integration in the postcolonial era [4]. The most effective bulwark against the resurgence of nationalist exclusivism is provided by a robust and democratic civil society [5]. Democratic grassroots civic activities can liberate society from its authoritarian past and offer democratic leadership for the future. To create a democratic culture and a strong international society that is a counterbalance to authoritarianism and dictatorship, civil society organisations (CSOs), both at the international and national levels, have been essential [6–8]. The first and second waves of democracy established the state's role in promoting democracy, whereas the third wave saw an expansion and intensification of civil society organisations' efforts to promote democracy in developing countries. However, studies have shown that many of these democracies are new and unstable, particularly in Africa. Furthermore, according to other analyses, the gains made during the earlier phase of Huntington's third democratic wave in Africa have recently slowed or halted [5, 9].

African democratic consolidation has received various debates on the involvement of international civil society, some of which can be contradictory. Some academics and political experts have asserted that national and international civil society have significantly contributed to the advancement of democratic culture [10]. They attributed the decline in liberal democratic culture to the unsubstantiated political preferences of some governments and the lack of room for an active civil society. Others said that the development of a democratic culture inside civil society was hindered by internal issues and lacked a unified voice. Such disintegration is attributable to the presence of surrogates of colonial regimes that have become part of the post-colonial government [2, 3, 11].

Some studies have acknowledged the importance of local and international civil society organisations in fostering a strong democratic culture, but many have expressed scepticism about how much this effect extends to liberal democracies in Africa, particularly Ghana [12]. This was brought on by Ghana's rising levels of political vigilantism, fraud in elections, corruption, and the hijacking of local civil societies by political elites. Nevertheless, regarding the consolidation of democracy, civil society and its myriad organisations appeared to have played a significant role in Ghana's democratic process [13]. Ghana was regarded as a "model of democracy in Africa" by academics and political commentators. Further consolidation of democracy is required as Ghana faces internal instability that threatens its fourth republic [14, 15].

However, other research argues that Ghana's democracy belies its name because it is still rife with corruption, despite the development of domestic civil society organisations and its multiparty political system. Hence a "democratic rupture," exists in Ghana and it revealed Ghana's democratic culture's weaknesses [16]. The International Civil Society Organization is an alternative to local civil society. What has been accomplished by international civil society organisations (ICSOs) to stop the corruption epidemic? The current study examined more closely the impact of the ICSO's role in combination with other mechanisms in fighting corruption. By discovering how international civil society assisted in the thorough development of a democratic culture with its greater capacity and facilities, this study critically investigated and filled the deficiencies related to the liberal democratic culture in Ghana. More crucially,

the current study addresses the temptation of assuming that a single strategy can address corruption by examining the role of ICSO and other variables using crisp-set Qualitative Comparative Analysis which create several configurations that are effective for anti-corruption.

## 1.2 The relevance of Civil Society Organization in the international system

Civil society has become relevant in both international and domestic politics. While nation-states remain the primary players in international politics, civil society is gaining significance [11]. Although post-World War II was characterised by the formation of several international organisations, their significance has not been well appreciated. The rise in economic, political, social, and cultural transactions between people, communities, and nations is paralleled by an increase in the number of international organisations. The proliferation of non-state players challenges and even undermines the "state-centric" notion of international politics, which is replaced with a "transnational" system characterised by increasingly complex interactions [17].

Based on the increasing number of non-state actors in the international political space, states are confronted by non-state entities that are gaining status and influence. To explain such trends, new theories of international relations, such as "complex interdependence" [18] by our colleagues [18], were developed. Civil society organisations are institutions founded by non-state players, or at least one side of which is not a state. There are numerous types of civil society organisations (CSOs), including transnational, donor-organised, donor-dominated, people's organisations, operational, advocacy, transnational social movements, quasi-governmental, and anti-governmental CSO. In the last decade of the twentieth century, their membership exceeded 23,000, and their efficacy in global politics was more significant in the twenty-first century [19]. They have become "vital players" in international policymaking. CSOs create and/or mobilise global networks by establishing transnational organisations, gathering information on local conditions through contacts around the world, alerting the global network of supporters to situations requiring attention, initiating emergency responses around the globe, and mobilising pressure from outside states. They participate in IGO conferences by mobilising transnational social movement organisations around IGO issues, constructing transnational social coalitions, raising new issues, supporting IGO development, addressing IGO meetings, submitting documents to meetings of governmental organisations, enhancing conference diplomacy skills, and increasing issue expertise. They facilitate interstate cooperation by preparing background papers and reports, educating delegates and state representatives to close technical gaps, serving as a third-party source of information, expanding policy options, facilitating agreements, and bringing delegates together in third-party fora [18, 19].

As a result of the accelerated globalisation process, transnational CSOs have become increasingly influential in determining the foreign policies of nation states [20]. Similar to their colleagues, who act at the national level and lobby in their nations, they lobby at the international and transnational levels. Human rights campaigners, gender activists, religious organisations, developmentalists, and indigenous peoples have invaded nation-state territories. As Brown notes, "as countries and sectors of world society have become increasingly interdependent, it has become commonplace for non-governmental groups representing similar communities in different countries to closely coordinate their policies and form (or reconstitute) themselves as international non-governmental organisations (INGOs)" contributing therefore to the democratisation process [21].

## 1.3 Anti-corruption strategy in Ghana through Civil Society Organization

There are various schools of thought regarding the role that CSOs play in fostering a democratic culture in Ghana. According to the first school of thought, Ghana's civil society helped

to ensure fair elections, supported policy changes, and encouraged civic engagement during the democratic process. As it grows, partisan politics undermine Ghana's democracy, and vigilantism-related violence, intimidation, and coercion hamper the electoral process. Ghana received a score of 40 on the Corruption Perceptions Index (CPI) as of September 2018, according to Transparency International [22]. Several academics and political analysts have argued that multiparty elections and political transition processes should not be the only evaluation variables for gauging a viable democratic culture because of Ghana's deepening corruption and lack of vigilance. To accurately reflect Ghana's "variety of democracy," additional variables should be included.

Previous studies examined the significance of civil society in democracies. For instance, in an online interactive discourse on how civil society influences democratic deliberation in a democratic culture, news consumption, political engagement, and social trust significantly mediate each other [23]. It examined how the media, civil society, and state political institutions have impacted the liberal democratic process. Several obstacles and problems must be overcome to maintain and further the democratic consolidation process. However, the direction of future free speech and anti-corruption campaigns should be a fundamental transformation in society toward a democratic culture. The four anti-corruption tactics that affected the absence of corruption were leadership, rules and regulations, training, and the threat of punishment. Additionally, it may be concluded that despite the forceful and energetic lobbying of international civil society for honesty, CSOs are unable to stop the recent wave of corruption and vigilantism due to a lack of accountability among state actors. However, adding other variables can provide configurations that make the role of the international civil society more impactful. While Ulhlin's study examined multiple variables on how civil society contributes to democracy in the case of Lativia, regression analysis does not provide causal complexities that show the interaction of these variables in a configuration. In addition, the challenge of correlation and regression analysis is that it creates a net additive problem [24–26]. Therefore, this study fills this gap in methodology by employing cs-QCA in civil society in Ghana, focusing on anticorruption strategies.

The objective of this study is to evaluate, from the assessment of CSOs, how many factors contribute to the development of democratic culture based on anti-corruption. Therefore, anti-corruption as an outcome is assumed to be impacted by the following conditions: the role of the International Civil Society (RICSO), freedom of speech, social trust, leadership, bureaucratic rules and regulations, fear of punishment, and training. The RICSO evaluates the impact of the three civil societies, Amnesty International (AI), Transparency International (TI), and the Global Organization of Parliamentarians against Corruption (GOPAC), which are present in Ghana. This study provides deeper knowledge of the contributions of international civil society groups, including their support for free and fair elections, observation and assessment of social and economic policies, and instruction in conflict management and prevention in Ghana. This will also contribute to knowledge on economic efficiency, equality, equity, and transparency in new democracies [27]. It discusses the implications of the roles that leadership, rules, and regulations, as well as fear of punishment, have in decreasing corruption. It suggests how ICSO should support civil society organizations in Ghana to shape national policy to advance accountability and openness.

## 2. Applying crisp-set QCA in measuring anti-corruption strategy

QCA is a research methodology and data analysis technique that seeks to merge case-oriented and variable-oriented approaches to identify configurations of conditions that account for an outcome. Concerning the quality of the analysis, QCA as a research strategy has a substantial

impact on the credibility of the results [28]. As a case-oriented approach, it seeks to gather in-depth information in many instances and grasp the complexity of cases. In the context of this study, the cases are civil society organisations that have assessed the conditions that account for anticorruption strategies. Set-theoretic techniques are one of the characteristics of QCA; they operate on the membership scores of the cases examined in set relations, creating subset relations while identifying the necessary and sufficient conditions. Employing set theory, intricately interconnected causal patterns are deconstructed in terms of equifinality, conjunctural causation, and asymmetry [29].

It has been argued to be applicable when there are theoretical grounds for the condition concerning the outcome. Some QCA scholars have argued that, even if a medium-N dataset is provided, researchers should avoid using QCA in the absence of specific assumptions on set relations. This study is based on democratisation theories that emphasise the role of civil society as part of the social forces for democratic changes. Hence, the theoretical argument supports the empirical argument. Similarly, if the focus of the study is on set relations as opposed to correlations, the use of QCA would be acceptable even if N is quite large [30]. Therefore, this study examines 264 CSO in Ghana which has a large N, since the focus is on set relations and especially on configurations that provide for Ghana's anti-corruption strategy. QCA has been developed in three forms, crisp-set QCA, fuzzy-set QCA, and multi-value variant set QCA [28].

## 3. Method

### 3.1 Crisp set QCA

Crisp-set QCA involves the use of binary coding for factors and outcomes. The goal of csQCA is to employ a strategy that creates a middle ground between case- and variable-oriented approaches. Using Boolean algebra, this method assumes binary outcomes and causes factors, or "crisp" sets, in the sense that membership in each is unambiguous. A specific example either exhibits set membership characteristics (and is therefore assigned a score of one) or does not (and is therefore assigned a score of zero). csQCA acknowledges that the presence and absence of conditions are meaningful for the outcome. The strength of csQCA is that it reduces complexity using a truth table and overcomes the net additive challenges of other methods such as correlation and regression analysis. Nevertheless, it does not attempt to replace those analysis but compliments the knowledge generated by statistical analysis [31].

### 3.2 Selection of cases

For QCA case selection, the principle involves selecting cases that are sufficiently similar to compare but differ sufficiently across conditions and outcomes. The CSO was selected based on its common focus on good governance. Hence, CSO that focused on other aspects, such as religion or trade, were excluded. They vary across the aspects of governance which they address; election monitoring, budget and accountability, political and economic rights, and anti-nepotism are significant areas of variation. The cases selected are 264 CSOs focusing on governance across the 216 local districts in Ghana and collected their assessments on the seven conditions and outcomes in 2021. The objective is to ensure a nation-wide coverage of civil societies in Ghana.

### 3.3 Research instrument

The study employed two research instruments to collect data from the participant in the study. The research instruments were semi-structure interview guide and survey design

questionnaire. The semi-structured interview guide adopted from some studies was employed as an instrument for collection of data for the study [32]. The guide contained open ended questions that covered relevant issues regarding the activities of ICSOs in the promotion of viable democratic culture in Ghana. Worthy of note were the questions designed to elicit vital information on what ICSOs were doing currently to underscore democratic culture in Ghana, the challenges they were facing and how they can successfully enhance cultivation of democratic culture in Ghana. Hence, CSOs evaluation of the role of ICSOs in Ghana was conducted. The guide was constructively criticized to eliminate ambiguous, non-specific, hypothetical, and misleading questions before the final administration. The survey design questionnaire consists of two parts to investigate the research model and formulated research questions three and four. The first part included questions about the demography of the participants and the second part included five constructs with twenty-five (25) questionnaire survey items in 5-point Likert Scale adapted from previous studies transformed for crisp set analysis [33].

## 3.4 Data collection procedure

Prior to the administration of questionnaire, the Cyprus International University Ethical Review Committee approved the study in accordance with the Declaration of Helsinki regarding research relating to human subjects in 2021. Interactions with the CSOs were established via emails, phone calls and a prior visit to their offices in the last quarter of the year 2021. These interactions were extremely important to consenting to participation. The product was a deeper understanding of the study phenomenon because the participants were able to speak freely and feel safe. Afterwards, each CSO represented by administrative officers and secretaries responded to the questionnaire items which begins with the clause on consent and confidentiality of respondents' information. The 264 responses on the survey questionnaire were analysed through crisp set [32].

## 3.5 Conditions

This study examines seven conditions that impact the anti-corruption strategy in Ghana among civil society organisations. This study examines seven conditions that impact the anti-corruption strategy in Ghana among civil societies. Some variables have been discussed independently of each other in several studies. This study brings them together to examine how the impact of anticorruption is different when certain conditions are present or absent.

*The role of the International Civil Society (RICSO)* refers to the level of demands and pressure mechanisms placed on government by the civil society organizations that operate across the globe. These organizations play a monitoring role in elections and other governmental decisions. Amnesty International (AI), Transparency International (TI), and the Global Organization of Parliamentarians against Corruption (GOPAC) are present in Ghana (GOPAC)'s roles have been assessed [32].

*Freedom of expression (FE)* points to the ability of individuals and organizations to question the conduct of governments and demand accountability. It involves the expression of *(dis)pleasure* towards any government or political action. The right to express agreement or disagreement is therefore assessed [33].

*Social trust (ST)* consists of the belief in the government and its apparatus to perform its function effectively with honesty and integrity. Therefore "faith in government" is examined in this context [34].

*Leadership (LD)* refers to the ability to lead and maintain a change toward a commitment to the standards, vision, and mission of good governance. A general assessment of the lead character of the government is examined [35].

*Rules and Regulations (RR)* consists of the standard operating procedures that are established for ministries and other government apparatus to fulfil their duties. The level at which adherence to these SOP mitigates corruption is evaluated [36].

*Fear of Punishment (FP)* points to the act of corruption carried out by government officials is expected to be met with appropriate consequences to deter others from engaging in any misconduct. The level at which the officials avoid corruption due to administrative, legal, and judiciary measure that will be taken against them is measured [37].

*Training (TR)* involves exercise directed towards ensuring that government officials, politicians as well as citizens are responsible in their conduct of national affairs. Examples are aptitude tests for civil servants, police officers training, and national service training [36].

*Anticorruption Strategy* (AC) represents the outcome which is the state of inhibition of all forms of corruption and malpractices in political offices, which contributes to the continuous development of democratic process and good governance [38].

### 3.6 Calibration

The cases were assigned membership scores based on seven factors: RICSO, freedom of expression, social trust, leadership, rules and regulations, fear of punishment, and training) and the outcome (anticorruption) through the evaluation of the 264 cases on good governance. Binary coding (1, 0) was used for explanatory conditions and outcomes. The presence of ROICS was coded as 1, whereas their absence was coded as 0. This was performed for all factors and conditions in each case. Grimes' study on the success of civil society in combating corruption [25] and Dixit's study on anticorruption institutions [26] provide the expectation that these conditions and outcomes have an interaction that defines an effective strategy. Therefore, the presence of anticorruption strategy is considered effective if the outcome receives 1 and ineffective if it is receives 0 based on the interaction of the factors [39].

## 4. Results

### 4.1 Analysis of necessary conditions for anticorruption strategy in Ghana

This section begins by analysing the necessary conditions for anticorruption in Ghana. In Table 1 below, necessary condition is a factor that is expected to appear in solution terms consistently. The implication is that if $X \geq Y$, therefore condition X is necessary for outcome Y. The conditions that are above 0.7 is considered necessary for the presence of an anticorruption strategy. In the table below, the Role of Civil Society (RICSO), Leadership, and fear of punishment have a consistency of 0.8 and therefore are expected to be in the solution terms that provide for anti-corruption. However, RC has the highest coverage of 0.8 compared to Leadership and Fear of Punishment which shows 0.667 and 0.400 respectively. Therefore, the Role of

**Table 1. Analysis of necessary conditions for the anticorruption strategy.**

| Conditions tested | Anticorruption in Ghana | |
| --- | --- | --- |
| | Consistency | Coverage |
| RICSO | **0.800** | **0.800** |
| FE | 0.600 | 0.429 |
| ST | 0.000 | 0.000 |
| LD | 0.800 | 0.667 |
| RR | 0.200 | 0.250 |
| FP | 0.800 | 0.400 |
| TR | 0.200 | 0.500 |

**Table 2. Truth table.**

| Conditions | | | | | | | Outcome | Consistency | | |
|---|---|---|---|---|---|---|---|---|---|---|
| RICSO | FE | ST | LD | RR | FP | TR | AC | raw consist. | PRI consist. | SYM consist |
| 1 | 0 | 0 | 0 | 0 | 1 | 0 | 1 | 1 | 1 | 1 |
| 1 | 1 | 1 | 0 | 1 | 1 | 0 | 0 | 0 | 0 | 0 |
| 1 | 1 | 0 | 1 | 0 | 1 | 0 | 1 | 1 | 1 | 1 |
| 1 | 1 | 0 | 1 | 1 | 1 | 0 | 1 | 1 | 1 | 1 |
| 1 | 1 | 0 | 1 | 0 | 0 | 1 | 1 | 1 | 1 | 1 |
| 0 | 1 | 0 | 0 | 0 | 1 | 0 | 0 | 0 | 0 | 0 |
| 0 | 1 | 0 | 0 | 1 | 1 | 0 | 0 | 0 | 0 | 0 |
| 0 | 0 | 0 | 0 | 0 | 1 | 0 | 0 | 0 | 0 | 0 |
| 0 | 0 | 0 | 1 | 0 | 1 | 0 | 0 | 0.5 | 0.5 | 0.5 |
| 0 | 1 | 1 | 1 | 1 | 1 | 1 | 0 | 0 | 0 | 0 |

Internatinoal Civil Society (RISCO) is necessary in the solutions that presents a causal path as a path to mitigating corruption in Ghana.

## 4.2 The truth table

Table 2 presented the Boolean truth table which displays all the possible configurations of conditions regarded as logical combinations before the process of logical minimization that simplifies the table into solution terms [40]. While 0 represents the absence of the outcome, 1 represents the presence of an outcome or condition. Ten possible combinations of conditions have been presented to account for the presence of an anticorruption strategy in Ghana. The solution shows that there is a need for minimization to remove redundant solutions where the presence or absence of a condition does not change the outcome.

## 4.3 Conditions for the presence of an anticorruption strategy in Ghana

After the minimisation process, the intermediate solution is examined and presented as the final combination of conditions [41]. The configuration that provides for the anti-corruption strategy in Ghana is presented in the Table 3, which shows the intermediate solution using a crisp-set analysis. The configuration affirms that the Role of Civil Society, Leadership, and fear of punishment are necessary conditions as they are present in the three combinations of conditions.

The first solution term reflects that the presence of the role of civil society, freedom of expression, leadership, and fear of punishment despite the absence of social trust and training will result in an effective anticorruption environment in Ghana. The second solution terms

**Table 3. The configuration for an effective anti-corruption strategy.**

| Configuration | Raw Coverage | Unique Coverage | Consistency |
|---|---|---|---|
| RISCO*FE*CS*LD*FP*~TR | 0.4 | 0.4 | 1 |
| RISCO*~FE*~ST*~LD*~RR*FP*~TR | 0.2 | 0.2 | 1 |
| RISCO*FE*~ST*LD*~RR*~FP*TR | 0.2 | 0.2 | 1 |

Solution Coverage: 0.8

Solution consistency: 1

* = combines

~ = absence of a condition

show that the presence of the Role of Civil Society and Fear of punishment in the absence of all other factors result in anti-corruption mechanisms in the country. The role of civil society, freedom of expression, and training despite the absence of all other conditions serves as the third solution term to anti-corruption. The role of Civil Society is therefore an insufficient condition but necessary and should be combined with all other factors as seen in the configuration. The overall consistency and coverage of the solution are 1 and 0.8 respectively. The first solution term explains 40%s percent of the cases in the configuration while the last two solution terms cover 20% of the cases in their respective configurations.

## 4.4 Conditions for the absence of an anticorruption strategy in Ghana

Table 4 shows the configurations that display the absence of an anti-corruption strategy in Ghana presented through an intermediate solution using a crisp-set analysis.

The first solution term shows that the presence of fear of punishment in the absence of the role of international civil society (RISCO), social trust, leadership, rules and regulations, and training results in the absence of an effective mechanism for addressing corruption in Ghana. Compared to the table on the presence of anticorruption in Ghana, fear of punishment is present with the RISCO (even in the absence of other conditions), there is a high level of anticorruption strategy.

The second solution terms show the presence of freedom of expression and fear of punishment in the absence of the role of international civil society, social trust, leadership, and training which explains the absence of an anticorruption strategy in Ghana. The third solution term shows that despite the role of international civil society, freedom of expression, social trust, rules and regulation and fear of punishment, the absence of leadership and training leads to the unyielding anticorruption strategy.

The last configuration in explaining the outcome of the ineffective anticorruption strategy in Ghana is the absence of the role of international civil society despite the presence of all other conditions. The role of international civil society is not a sufficient condition for the absence of a strategy in fighting corruption and the presence of all other conditions is sufficient as well to assert an effective anticorruption strategy in Ghana. Both explanations show that no one condition is sufficient to explain the case of anticorruption in Ghana but rather a combination of conditions interacting with one another to produce the outcome.

The overall consistency and coverage of the solution are 1 and 0.8333, respectively. The first and second solution terms explain 33% of the cases in the configuration while the other two solution terms cover 16% of the cases in their respective configurations.

**Table 4. The configuration for ineffective anti-corruption strategy.**

| Configuration | Raw Coverage | Unique Coverage | Consistency |
|---|---|---|---|
| ~RISCO*~ST*~LD*~RR*FP*~TR | 0.333333 | 0.166667 | 1 |
| ~RISCO*FE*~ST*~LD*FP*~TR | 0.333333 | 0.166667 | 1 |
| RISCO*FE*ST*~LD*RR*FP*~TR | 0.166667 | 0.166667 | 1 |
| ~RISCO*FE*ST*LD*RR*FP*TR | 0.166667 | 0.166667 | 1 |

Solution Coverage: 0.83333

Solution Consistency: 1

* = combines with

~ = absence of condition

## 5. Discussion and conclusion

Understanding the pathways to mitigating corruption is important for the continuous development of democracy across the world. According to the Democratic Index of the Economist Intelligence Unit (EIU), African countries rank lowest in the democratisation process. Democratisation includes political, economic, and social elements that ensure that the government is for the people. However, the absence of certain conditions has led to a continuous backsliding in democracies. Political corruption has been identified as a major factor affecting the democratisation process in Africa. This study examined Ghana as the country that first began its democratic path through independence. The objective of this study is to examine the anti-corruption strategy in Ghana to understand the interactions among the factors that contribute to mitigating corruption.

This study examined the impact of seven anticorruption conditions that support the promotion of democracy in Ghana. The role of international civil society (RICC) has been examined in studies that emphasise the role of non-governmental international actors in promoting democracy. The promotion of human rights, gender equality, and seeking accountability from governments across the globe for the treatment of their citizens and resources has been central to most international civil societies [6]. Recent studies have focused on regular elections and how changes in the government can mitigate corruption. The evidence shows that a change in government does not fight corruption; rather, it provides more opportunities for the next government to take its share of national resources to its advantage. Other domestic factors, such as the level of freedom of expression, have significantly determined the level at which citizens can report cases of corruption without being hunted by the ruling or incumbent government. These studies focused on univariate analysis, which involved individual analysis of factors [7, 8].

The study argues that univariate analysis, as seen in previous studies, does not provide sufficient analysis to understand anticorruption strategies. Other studies used multivariate approaches [24–26]. A multivariate approach provides a more in-depth analysis of the impact of anti-corruption by presenting several configurations that lead to the outcome. This study makes a significant contribution as it examines seven factors that mitigate corruption in Ghana. The Role of International Civil Society (RC), freedom of expression (FE), social trust (ST), leadership (LD), Rules and Regulations (RR), Fear of Punishment (FP), and training (TR) are conditions that contribute to anticorruption.

This study examined the interaction of variables to determine the presence or absence of an anti-corruption strategy. The crips set a qualitative comparative analysis method that has been employed in many studies to examine the different conditions for an outcome because of its multiple causal path solutions [42]. Similarly, this study uses csQCA to explore configurations for effective anti-corruption strategies and configurations to avoid, because they lead to the ineffectiveness of the strategy. Using primary data collected from 264 civil society organisations on the identified conditions that contribute to anticorruption, the results show the significant role of international civil society in countering corruption in Ghana.

The results of this study reaffirm the arguments of scholars of complex independence in international politics and present the causal complexities suggested by QCA which is the first attempt to present this duo: theoretical and methodological contributions [17, 18]. To provide solutions to Ghana's performance in the Corruption Perception Index, most studies are limited because they only explain the correlation between civil society and democracy without recommending or proposing the casual path or conditions that must be combined as recipes or configurations to improve anticorruption and lead to democratisation [22, 43]. This study shows a different recipe to the anticorruption strategy rather than only arguing for civil society

participation in Ghana. The first set of solution terms in this study shows that the presence of international civil society in combination with freedom of expression, good leadership, and fear of punishment results in an effective anticorruption environment in Ghana. Another solution term also showed that the presence of the Role of International Civil Society combined with fear of punishment results in an effective anti-corruption path in the country. Freedom of expression and training, in addition to the role of civil society despite the absence of all other conditions, is the third solution term that provides an effective anti-corruption recipe in Ghana.

However, some configurations do not contribute to the effectiveness of the anti-corruption mechanism despite the presence of certain conditions, and these configurations must be avoided. The absence of the role of civil society, social trust, leadership, rules and regulations, and training results in a lack of anti-corruption strategies in Ghana. Furthermore, the absence of a civil society, social trust, leadership, and training does not provide an effective anticorruption environment in Ghana. Finally, despite the role of civil society, freedom of expression, social trust, rules and regulations, and fear of punishment, these do not stand as anticorruption strategies without leadership and training.

This study recommends that pulling several anti-corruption strategies together should not be the goal of decision-makers and policymakers in Ghana. The combination of the factors explained in this study must be considered to execute an effective anticorruption strategy. In addition, international civil society plays a crucial role in contributing to the anticorruption strategy in Ghana. Hence, the significant impact of Amnesty International (AI), Transparency International (TI), and the Global Organization of Parliamentarians against Corruption (GOPAC) must be acknowledged because they seek accountability from the government and private sector while supporting local civil society in gaining more democratic space in Ghana. This study recommends that local civil society understudy the international civil society's strategy of getting government officials and politicians to say yes to becoming more committed to and accountable.

## Supporting information

**S1 File.**
(DOCX)

## Acknowledgments

The authors would like to thank Emmanuel Oluwatosin Adewusi for his helpful suggestions during the writing process.

## Author Contributions

**Conceptualization:** Ebenezer Kurtis Graham, Özker Kocadal.

**Formal analysis:** Ebenezer Kurtis Graham.

**Methodology:** Ebenezer Kurtis Graham, Özker Kocadal.

**Writing – review & editing:** Özker Kocadal.

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
