## [Decision Letter · Decision Letter 0]

20 Apr 2023

PONE-D-22-34363The role International Civil Society Organizations in democratization: A crisp-set QCA approach to Anti-corruption in Ghana.PLOS ONE

Dear Dr. Graham,

Thank you for submitting your manuscript to PLOS ONE. After careful consideration, we feel that it has merit but does not fully meet PLOS ONE’s publication criteria as it currently stands. Therefore, we invite you to submit a revised version of the manuscript that addresses the points raised during the review process.

Although the topic is intriguing and the research covers it in an academically relevant manner, a major revision is required to re-consider the paper for publication. Both reviewers point out structural problems, which the author/s need to take seriously into account. Both of the critical opinions agree that the literature review must be revisited and broadened. Rev 2 also underlines that the introduction is too long and repetitive, as well as there are several considerations to tackle about methodology, or better to say, the presentation of the methodology used for the research. I stress upon the requisite of a critical revision of the original manuscript therefore.

We look forward to receiving your revised manuscript.

Kind regards,

István Tarrósy, PhD

Academic Editor

PLOS ONE

Journal Requirements:

4. We note you have included a table to which you do not refer in the text of your manuscript. Please ensure that you refer to Tables 1, 2, 3 and 4 in your text; if accepted, production will need this reference to link the reader to the Table.

Reviewers' comments:

Reviewer's Responses to Questions

**Comments to the Author**

1. Is the manuscript technically sound, and do the data support the conclusions?

Reviewer #1: Yes

Reviewer #2: Partly

2. Has the statistical analysis been performed appropriately and rigorously? 

Reviewer #1: Yes

Reviewer #2: I Don't Know

3. Have the authors made all data underlying the findings in their manuscript fully available?

Reviewer #1: Yes

Reviewer #2: Yes

4. Is the manuscript presented in an intelligible fashion and written in standard English?

Reviewer #1: Yes

Reviewer #2: No

5. Review Comments to the Author

Reviewer #1: The topic of is interesting and relevant. The elaboration and the applied methodology are accurate, however the goal(s) and main questions/hypotheses should be explained in more details. The scope of bibliographic sources could be broadened as papers in Q1-Q2 journals should be based on at least 50 sources. It would be interesting to add a brief international comparison in this field at least with other African countries, like Nigeria in which combatting against corruption is a core issue as well. In case of each figure and table the sources of data/information must be indicated, even if the derive from the direct, primary research of the authors. The chapter of discussion of results and conclusion should be divided to two separate chapters, like Discussion, and Conclusion and suggestions. As for the latter, it can be extended with ideas for the applicability of the new results and findings. After the suggested improvements were made, the paper can be recommended to be accepted and published.

Reviewer #2: The paper works on a very relevant topic, namely the role of international (and local) civil society organizations in controlling corruption and, thus, contributing to democratisation in Ghana. However, publishing the article would be recommended only after major revisions.

The introduction is way too long, stuffed with information not necessarily relevant or many times repeated. Moreover, it misses to summarize the structure of the whole paper which could help the reader to focus on interested part, and it is a comme-il-faut custom for any introduction. Overall, it should be compressed and more straight-to-the-point.

It is also not exactly clear on many of the initial pages what exact role or actions of international and/or local (?) organizations are examined. It is suspected after some pages that local organizations were asked about their perception of the presence and actions of the big three international civil organizations, but it is still blurry. Furthermore, it is also not well-explained or justified by references on literature or other arguments why the given seven conditions were selected. Why not more? Why not less?

Section 2 should not be a separate section. It can simply go under Section 3 (Methodology). Nevertheless, the methodological choice does not seem justified. It is rather blurry and hardly digestible. On page 5 the authors write: „In the context of this study, the cases are civil society organizations that have assessed the conditions that account for anticorruption strategies.” This is a misinterpretation of cases or case studies. Meanwhile a case study is rather “best defined as an in-depth study of a single unit (a relatively bounded phenomenon) where the scholar’s aim is to elucidate features of a larger class of similar phenomena” (Gerring, 2004, p.341). So, in this sense these local CSOs are not cases, rather respondents or observations and this, with the large sample size, would make regression analysis more preferable. Other nuances related to the research instrument (questionnaire) are also problematic, such as why it is important to collect data on demography of respondents, or how the seven conditions were distributed to 25 questions (why not 21, or which conditions was probed by more questions and how this affected the results) in the second part of the questionnaire, or how 5-point Likert scale is adapted to the 0-1 character of QCA.

When presenting the results, the authors should consider that not all readers are familiar with their chosen methodology. Therefore, it should be explained what certain results and expressions mean and indicate. Such as, what “Consistency” and “Coverage” stand for and how these numbers were produced. So, the reader can easily get lost in the interpretation of the results. Furthermore, these results indicate interesting, sometimes surprising findings which should be more discussed and reflect on, in comparison with the literature and theory. One example for this is that, according to the authors, „social trust” does not play a significant role in anticorruption…and relatedly in democratisation. This seems to be a contradiction and would deserve more scholarly interpretation.

Beyond, Section 5 (Discussion and Conclusions) misses answering the „So what?” question and uses confusing argumentation, such as, in the last paragraph: „This study recommends that pulling several anti-corruption strategies together should not be the goal of decision-makers and policymakers in Ghana. The combination of the factors explained in this study must be considered to execute an effective anticorruption strategy.” These two sentences after each other are hardly understandable. The final recommendation of the study is that local civic organizations should learn from the practices of international ones to make politicians more accountable. One problem with this is that there is nothing new in this, and a second, more serious one is that the authors ignore the difference between the capacities/power of big internationals and small locals.

Finally, in many places the text contains typos, inconsistent usage of acronyms and missing references.

6. PLOS authors have the option to publish the peer review history of their article (what does this mean?). If published, this will include your full peer review and any attached files.

Reviewer #1: No

Reviewer #2: No

---

## [Author Response · Author response to Decision Letter 0]

16 Aug 2023

Response to Reviewer 1 

Thank you for your comments for improving on our research.

The objective has been rewritten in the new introduction with more details of the research questions. The scope of the bibliography has been broadened as sources for the manuscript have been increased to 53 as recommended by the reviewer. In the discussion, a comparative discussion was made concerning Nigeria and Kenya’s anti-corruption strategy. The discussion stands alone, while the conclusion and recommendations pointing to the applicability of the findings are the last sections.

Response to Reviewer 2

Thank you for your comments on improving our study.

Firstly, the structural problems identified in the previous version have been addressed. The manuscript introduction has been improved, addressing corruption and examining possible anti-corruption strategies. The previous introduction roamed democracy; as you noted unnecessary to roam around that. However, the current version quickly touches upon democracy only in connection to corruption. The structure of the paper has also been included in the last paragraph of the article. However, due to the broadening of the literature review, as suggested by Reviewer 1, the length of the introduction has only been shortened reasonably.

 It identified the problems in the case of Ghana, examined the literature that suggested different conditions, and highlighted the limitation of regression analysis. It concludes by stating the objective, the research questions, and the hypothesis. This resolves the length of the introduction and the repetitions in the introduction.

The connection of CSO and ICSO has been highlighted in the introduction.“Some studies suggest that International Civil Society Organizations (ICSOs) significantly strengthen local CSOs to overcome challenges by effectively mobilizing skills and financial resources. The CSO receives technical support from ICSO when facing challenges that limit its operation. The ICSO supports the CSO with funding, advocacy on issues being suppressed by the government, technical advice, and networking; it connects the CSO to a network that can provide other needed support for effectiveness. This study examined the CSO evaluation of the ICSO such as Transparency International, Amnesty International, and the Global Organization of Parliamentarians against Corruption (GOPAC), which supports CSO and pushes the government to address corruption”.

The seven conditions have been chosen based on the large number of respondents thus, the more the cases the more the respondents. “Seven conditions were chosen in compliment to the size of the population examined. According to Ragin, Schneider, and Wagemann, when the population is large, more conditions are needed to provide the diversity of explanatory factors that would bring about more configuration. Since the population size is 264, more factors are needed for the analysis”.

Your suggestion to change cases to respondents has been applied to the study as suggested since the study's main objective is to identify configurations. However, regression analysis does not provide the interaction among conditions to bring out an outcome, i.e., configuration. Hence, csQCA has been maintained. 

The data was collected based on demography because domestic CSO in Ghana has faced different challenges based on location. For example, the Northern CSO faced insecurity due to the corrupt practice of politicians who have failed to pursue infrastructure that deter bandits and nomads from attacking the people. While in the South, insecurity is based on natural disasters due to embezzlement of public funds. Other CSOs also experience the impact of corruption differently. Therefore, demography is a justified pattern of data collection rather than the capacity of the CSO because it represents the strategies CSOs suggested for anti-corruption by all-district.

Pappas and Woodside stated the feasibility of adopting the Likert scale into QCA. The Likert scale was adapted to csQCA using the STATA version 13 to generate variables to calculate the average of all questions that measure each variable. Following Parker’s study on analyzing criminology through csQCA, which used the arithmetic mean as the cut-off point when applying the Likert scale, our analysis also used the mean for the cut-off point to set membership score. Membership in conditions was calibrated as 0 for non-membership in condition/variable and 1 for the membership in the condition/variable according to the following cut-off; anti-corruption strategy 3.67, training 4.0, fear of punishment 3.67, rule and regulations 4.0, leadership 3.67, social trust 3.75, Freedom of Expression 3.67 and RICSO 4.0. Grimes’ study on the success of civil society in combating corruption and Dixit’s study on anti-corruption institutions provide the expectation that these conditions and outcomes have an interaction that defines an effective strategy. Therefore, an anti-corruption strategy is considered adequate if the outcome receives 1 and ineffective if it receives 0 based on the interaction of the factors. The necessary condition analysis, the truth table, and the configurations were conducted using QCA version 3.1b software.

The presentation of the methodology has been rewritten to highlight how we arrived at our conclusion. The missing point was how the Likert scale was converted and used for csQCA, which has been explained in the methodology section following best practices in the literature. STATA generate variable command was used to get the average score of each respondent to the questions of the variables, which was calibrated based on cut-off point into binary coding. 

Consistency and Coverage have been explained, i.e., consistency is the level of similar composition brought about the same result while Coverage is the percentage of the cases, in this case, respondents, covered by the solution. 

Regarding social trust, in the first configuration of an effective anti-corruption strategy, social trust was present (It was written as CS instead of ST, leading to misinterpretation); hence social trust is a significant factor present in 40 per cent of the cases analyzed. This has been corrected in the explanation of Table 3 and reflected in the explanation.

Discussion and Conclusion in the previous manuscript have been split into two sections; 1) Discussion and 2) Conclusion and Recommendation. These sections have been written carefully to engage readers that might not be conversant with csQCA.

The following paragraph below has been added to address the “So what” question. 

While CSO continues to look outward for support from ICSO, this study recommends that they continue to seek to strengthen the domestic mechanisms; freedom of expression, social trust, leadership, social trust, rules, regulation, and training for effective anti-corruption strategy, and getting government officials and politicians to say yes to becoming more committed to being accountable rather than seeing them as a tool used by developed countries. This conclusion is arrived at because ICSO cannot function without some domestic mechanisms being put in place. In this case, therefore, leadership against corruption and the fear of punishment for all corrupt conduct is activated.

Typographical errors have been addressed, as well as inconsistent acronyms, especially in the case of RICSO. Lastly, additional references have been added.

---

## [Decision Letter · Decision Letter 1]

29 Aug 2023

The role of International Civil Society Organizations in democratization: A crisp-set QCA approach to Anti-corruption in Ghana.

PONE-D-22-34363R1

Dear Dr. Graham,

We’re pleased to inform you that your manuscript has been judged scientifically suitable for publication and will be formally accepted for publication once it meets all outstanding technical requirements.

Kind regards,

István Tarrósy, PhD

Academic Editor

PLOS ONE

Additional Editor Comments (optional):

Reviewers' comments:

Reviewer's Responses to Questions

**Comments to the Author**

1. If the authors have adequately addressed your comments raised in a previous round of review and you feel that this manuscript is now acceptable for publication, you may indicate that here to bypass the “Comments to the Author” section, enter your conflict of interest statement in the “Confidential to Editor” section, and submit your "Accept" recommendation.

Reviewer #1: All comments have been addressed

Reviewer #2: All comments have been addressed

2. Is the manuscript technically sound, and do the data support the conclusions?

Reviewer #1: Yes

Reviewer #2: Yes

3. Has the statistical analysis been performed appropriately and rigorously? 

Reviewer #1: Yes

Reviewer #2: Yes

4. Have the authors made all data underlying the findings in their manuscript fully available?

Reviewer #1: Yes

Reviewer #2: Yes

5. Is the manuscript presented in an intelligible fashion and written in standard English?

Reviewer #1: Yes

Reviewer #2: Yes

6. Review Comments to the Author

Reviewer #1: The Author(s) made all corrctions and improvement wich the Reviewer recommended in the first reviewing round. Therefore, the paper can be accepted for publication.

Reviewer #2: This Revised version is much-much better than the original one. It is visible that the authors took into consideration and worked thoroughly on all the recommendations and comments. They convincingly explain why they chose the certain methodology, factors, etc. The structure is also much better and at the end of the paper they give a valuable contribution as to the literature and to policy-making issues.

7. PLOS authors have the option to publish the peer review history of their article (what does this mean?). If published, this will include your full peer review and any attached files.

Reviewer #1: No

Reviewer #2: No

---

## [Editor Report · Acceptance letter]

7 Nov 2023

PONE-D-22-34363R1 

The role of International Civil Society Organizations in democratization: A crisp-set QCA approach to Anti-corruption in Ghana. 

Dear Dr. Graham:

I'm pleased to inform you that your manuscript has been deemed suitable for publication in PLOS ONE. Congratulations! Your manuscript is now with our production department. 

Kind regards, 

on behalf of

Dr. István Tarrósy 

Academic Editor

PLOS ONE